# Pickering Emulsion Stabilized by Fish Myofibrillar Proteins Modified with Tannic Acid, as Influenced by Different Drying Methods

**DOI:** 10.3390/foods12071556

**Published:** 2023-04-06

**Authors:** Umesh Patil, Saqib Gulzar, Lukai Ma, Bin Zhang, Soottawat Benjakul

**Affiliations:** 1International Center of Excellence in Seafood Science and Innovation, Faculty of Agro-Industry, Prince of Songkla University, Hat Yai 90110, Thailand; umesh.p@psu.ac.th; 2Department of Food Technology, Engineering and Science, University of Lleida—Agrotecnio CERCA Center, Av. Alcalde Rovira Roure 191, 25198 Lleida, Spain; sgwani@hotmail.com; 3Guangdong Provincial Key Laboratory of Lingnan Specialty Food Science and Technology, College of Light Industry and Food, Zhongkai University of Agriculture and Engineering, Guangzhou 510408, China; malukai@zhku.edu.cn; 4Academy of Contemporary Agricultural Engineering Innovations, Zhongkai University of Agriculture and Engineering, Guangzhou 510408, China; 5Key Laboratory of Health Risk Factors for Seafood of Zhejiang Province, College of Food Science and Pharmacy, Zhejiang Ocean University, Zhoushan 316022, China; zhangbin@zjou.edu.cn; 6Department of Food and Nutrition, Kyung Hee University, Seoul 02447, Republic of Korea

**Keywords:** Pickering emulsion, protein–polyphenol interaction, emulsion stability, storage, rheological studies

## Abstract

A novel food-grade, particles-based Pickering emulsion (PE) was prepared from a marine source. Yellow stripe trevally is an under-utilized species. The use of its muscle protein as solid food-grade particles for the preparation of a Pickering emulsion can be a potential means of obtaining the natural nutritive emulsifier/stabilizer. Fish myofibrillar proteins (FMP) were modified with tannic acid (TA) at varying concentrations (0.125, 0.25, and 0.5%) followed by freeze-drying (FD) or spray-drying (SD). Physicochemical characteristics and emulsifying properties of obtained FMP-TA complexed particles were assessed for structural changes and oil-in-water emulsion stabilization. The addition of TA caused a reduction in surface hydrophobicity and total sulfhydryl content values for either FD-FMP or SD-FMP. Conversely, disulfide bond content was significantly increased, particularly when TA at 0.5% was used (*p* < 0.05). FTIR, spectrofluorometer, and the protein pattern also confirmed the cross-linking between FMP and TA. SD-FMP modified with 0.5% TA (SD-FMP-0.5TA) rendered the highest emulsifying stability index but had a lowered emulsifying activity index (*p* < 0.05). Confocal microscopic images, droplet size, and rheological properties revealed that a SD-FMP-0.5TA-stabilized emulsion had higher stability after 45 days of storage than an FD-FMP-0.5TA-stabilized emulsion. Therefore, the SD-FMP-0.5TA complex could be used as a potential food-grade stabilizer/emulsifier for PE with enhanced emulsifying properties.

## 1. Introduction

An emulsion consists of two immiscible liquids (usually oil and water) held in suspension by additives so-called emulsifiers/stabilizers. Recently, the Pickering emulsion (PE) has drawn attention due to its high emulsion stability. The long-term emulsion stability of PE is achieved via the irreversible adsorption of solid particles with amphiphilic characteristics on the oil droplet surface and the development of film around oil droplets [1]. Most research is concentrated mainly on rigid particles, e.g., metal oxides, calcium carbonate, and silica to prepare PE with various applications [2]. Nevertheless, inorganic ions have restrictions in the nutraceutical, pharmaceutical, and food sectors. Therefore, the utilization of natural biopolymer as an effective emulsifier/stabilizer has become more important. PE has been prepared by using polysaccharide-based particles involving modified small granular starches [3], corn starch nanoparticles [4], chitosan/cellulose complexed particles [5], protein-based microgels or particles including soy protein, whey protein, pea protein, fish gelatin, and polysaccharide–protein complex, e.g., zein-pectin complexed nanoparticles [6,7,8]. Among these, polysaccharide–protein or protein-based particles with an amphiphilic nature are the most effective in the stabilization of emulsion by lowering the interfacial tension and preventing coalescence, creaming, and Ostwald ripening in PE [9].

Yellow stripe trevally (*Selaroides leptolepis*) is a common dark-fleshed fish found in Southern Thailand. This is an under-utilized species that is generally known as a low-value marine resource with minimal market value, and it is very susceptible to spoilage. For human consumption, fish meat has served as an excellent source of proteins with essential amino acids. The use of fish myofibrillar proteins to develop PE offers a novel approach for alternative raw materials and also minimizes waste. Moreover, the long-term emulsion stability of PE can be achieved via the adsorption of novel fish myofibrillar proteins–tannic acid (FMP-TA) complexed solid particles on the oil droplet surface and the development of film around oil droplets. It is known that interactions between proteins and polyphenols affect colloidal particle behavior at interfaces [10]. Naturally occurring polyphenols such as tannic acid (TA) are widely distributed in abundance. TA has numerous digalloyl ester groups that create hydrogen bonds with protein [11]. Therefore, protein–TA interactions have gained attention for the enhancement of interfacial properties of proteins including sodium caseinate [12], zein [13], and gelatin [13]. However, information on the interaction of FMP with TA and the impact of drying methods to modify the protein structure of generated particles is unavailable. Therefore, the present study aimed to examine the interactions of FMP with TA at varying concentrations, followed by drying with different methods (freeze-drying and spray-drying), and to characterize the FMP-TA complex as an emulsifier/stabilizer. The morphology, droplet size, and rheological property of the emulsion prepared by the FMP-TA complex were also determined before and after storage for 45 days.

## 2. Materials and Methods

### 2.1. Chemicals and Materials

All the chemicals used were of analytical grade and procured from Sigma (St. Louis, MO, USA). Deceased yellow stripe trevally (*S. leptolepis*) (1 day after capture), the lives of which were ended by the vendor, were purchased from a local market in Hat Yai.

### 2.2. Preparation of Fish Myofibrillar Proteins

Yellow stripe trevally was cleaned with tap water, filleted, and manually chopped. Cold distilled water (DI, three volumes) was added to fish mince and homogenized at 13,000 rpm for 2 min. After that, the mixture was filtered by using a nylon cloth. The washing process was repeated twice to remove the blood and concentrate myofibrillar proteins. Then, washed mince was centrifuged at 3000× *g* for 15 min at 4 °C by using a refrigerated centrifuge (Beckman Coulter, Allegra™ centrifuge, Palo Alto, CA, USA). Thereafter, the obtained pellet was kept at a refrigerated temperature (4 °C) until further use.

### 2.3. Preparation of Fish Myofibrillar Protein–Tannic Acid (FMP-TA) Complexed Particles

The obtained fish myofibrillar protein (FMP) pellet was suspended in DI and adjusted to pH 10.5 with 2 M NaOH. Protein content was determined via the biuret method by using BSA as standard [14]. Tannic acid (TA) at various concentrations (0.125%, 0.25%, and 0.5%) was added to the FMP solution and stirred for 1 h at 4 °C. The mixture was then freeze-dried by using a SCANVAC CoolSafe™ freeze-dryer (CoolSafe 55, ScanLaf A/S, Lynge, Denmark) or spray-dried by using a laboratory scale spray-dryer (LabPlant Ltd., LabPlant SD-05, Huddersfield, UK) at a feed rate of 5 mL min^−1^. The inlet and outlet temperatures were kept at 180 ± 2 °C and 105 ± 2 °C, respectively. The airflow rate was fixed at 4.5 ms^-1^. The control sample was prepared in the same manner, except that TA was excluded. The obtained samples were placed in a zip-lock bag and stored at −40 °C until further analyses.

#### 2.3.1. Surface Hydrophobicity (S_o_), Total Sulfhydryl Content (TSC), and Disulfide Bond Content (DSBC)

The methods described by Benjakul et al. [15] were adopted for the determination of S_o_, TSC, and DSBC of freeze-dried (FD) and spray-dried (SD) FMP samples modified with TA at varying concentrations.

#### 2.3.2. Protein Structures

Alternation in protein structures of FD-FMP and SD-FMP modified with TA at varying concentrations was elucidated with the aid of a Fourier transform infrared (FTIR) spectrophotometer and spectrofluorometer, as per the methods of Li et al. [16]. To obtain quantitative information of the secondary structure of proteins, deconvolution of the amide I band (1700–1600 cm^−1^) was performed by using PeakFit V4.12 software (Systat Software Inc., San Jose, CA, USA).

#### 2.3.3. Protein Patterns

Protein patterns of FD-FMP and SD-FMP modified with TA at varying concentrations were determined via SDS-PAGE [17]. In brief, a 0.5 g sample was mixed with 10 mL of 5% SDS, and the mixture was heated at 95 °C for 30 min with subsequent centrifugation at 8000× *g* for 15 min. The biuret method was used to measure the protein content of the solution [14]. The samples were then added to a sample buffer containing 0.05% bromophenol blue, 10% glycerol, and 2% SDS in Tris-HCl (0.5 M, pH 6.8) without and with 5% β-mercaptoethanol, representing a non-reducing and reducing condition, respectively. Finally, 15 μg of protein samples were loaded on freshly prepared polyacrylamide gel (12% running gel; 4% stacking gel). Gels were separated at 15 mA per gel, followed by staining and destaining. Protein standards were also applied, and the molecular weight of protein bands in the samples was then calculated.

### 2.4. Preparation of Emulsion

FD-FMP and SD-FMP samples modified with TA at varying concentrations were firstly dispersed in DI containing 0.02% sodium azide (as an antimicrobial agent) to obtain 1% protein. The mixture was agitated overnight at a refrigerated temperature (4 °C) by using a magnetic stirrer. Emulsions were obtained by mixing soybean oil and prepared FD-FMP-TA or SD-FMP-TA suspension at a ratio of 1:2 (*v*/*v*). A high-speed homogenizer (IKA, Labortechnik, Selangor, Malaysia) was used to homogenize the mixtures (11,000 rpm for 2 min). The prepared emulsions were used for further study.

#### Emulsifying Properties

Emulsifying properties of prepared emulsions were measured in terms of emulsifying activity index (EAI) and emulsion stability index (ESI), following the modified method of Pearce and Kinsella [18]. EAI and ESI calculations were performed as follows:(1)$$EAI(m^{2}/g)=\frac{2\times 2.302\times A_{0}\times DF}{C\times L\times (1-\emptyset )\times 10000}$$
(2)$$ESI(min)=\frac{A_{0}\times \Delta t}{A_{0}-A_{30}}$$
where A_0_ and A_30_ are the absorbance taken at 0 and 30 min, respectively. DF is the dilution factor; C refers to the initial protein concentration (g/mL); L is the path length (m); ∅ is the oil volume fraction; and Δt = 30 min.

### 2.5. Characterization and Storage Stability of the Emulsion

A sample showing the highest ESI was used for further study. The emulsion was prepared by using FD-FMP-TA or SD-FMP-TA, and the selected concentration was stored at 28 ± 2 °C for 45 days. Freshly prepared and 45-day-stored emulsions were determined for microstructure, droplet size (coalescence and flocculation), and rheological properties.

#### 2.5.1. Confocal Laser Scanning Microscopy (CLSM)

The microscopic structures of samples were visualized via CLSM (Model FV300; Olympus, Tokyo, Japan) by using Nile Blue A dye following the procedure of Patil and Benjakul [19]. The CLMS in the fluorescence mode (emission wavelength: 630 nm; excitation wavelength: 533 nm) was used. For lipid analysis, a Helium-Neon Red laser (HeNe-R) was applied with a magnification of 400×.

#### 2.5.2. Droplet Size

The droplet size of emulsions was measured by using a laser droplet size analyzer (ANALYSETTE 22 NanoTec, FRITSCH, Idar-Oberstein, Germany) [20]. Prior to analysis, the sample was mixed with SDS at 1:2 (*v*/*v*) for the separation of flocculated oil droplets. The surface-weighted mean particle diameter (*d*_32_) and the volume-weighted mean particle diameter (*d*_43_) of the emulsion droplets were determined [21].

##### Flocculation and Coalescence

Emulsion samples were diluted with DI in the absence and presence of SDS. The flocculation factor (*F_f_*) and coalescence index (*C_i_*) were calculated by using the following equations [22]:(3)$$F_{i}=\frac{d_{43}-SDS}{d_{43}+SDS}$$
(4)$$Ci=\frac{\left(d_{43}+SDS,t-d_{43}+SDS,i\right)}{d_{43}+SDS,in}\times 100$$
where *d*_43_ − SDS and *d*_43_ + SDS are the volume-weighted mean particle diameters of the emulsion droplets in the absence and presence of SDS, respectively; *d*_43_ + SDS,i and *d*_43_ + SDS,t are the volume-weighted mean particle diameters of the emulsion droplets in the presence of SDS at the initial day (day 0) and the designated storage (45 days). The analysis was performed at ambient temperature (28–30 °C).

#### 2.5.3. Rheology

The initial (day 0) and stored samples (day 45) were evaluated for rheological properties by employing a rheometer (RheoStress RS 1, HAAKE, Karlsruhe, Germany) with parallel geometry (60 mm diameter, and 1 mm gap), as tailored by Patil and Benjakul [22]. The linear viscoelastic range was determined by using a strain sweep from 0.1% to 100% at a fixed frequency (1.0 Hz). Frequency sweep was performed with a constant strain (0.5%) throughout the linear region and over a selected frequency range (0.1–10 Hz). The storage modulus (G′) and loss modulus (G″) were determined as a function of frequency at 25 °C. Viscosity was obtained by using an increasing shear rate of 1–100 s^−1^ within 2 min.

### 2.6. Statistical Analysis

A completely randomized design (CRD) was used for the entire study. Three lots of samples were used to conduct the experiments in triplicate. The *t*-test was performed to compare the pairs by using the Statistical Package for Social Science (SPSS 23.0 for windows, SPSS Inc., Chicago, IL, USA) [23]. Moreover, the effects of two independent factors (concentrations of tannic acid and types of drying method) on the selected dependent variables (analytical parameter) were analyzed by using two-way ANOVA (GraphPad Software, San Diego, CA, USA) followed by Dunnett’s multiple comparison.

## 3. Results

### 3.1. Surface Hydrophobicity (S_o_), Total Sulfhydryl Content (TSC), and Disulfide Bond Content (DSBC)

S_o_ values of FD-FMP and SD-FMP modified with TA at varying concentrations are tabulated in Table 1. S_o_ reflects vital structural properties related to the change in protein conformation that governs the functional properties of proteins [24]. All SD-FMP-TA showed significantly higher S_o_ values compared to FD-FMP-TA samples, regardless of the TA concentration used (*p* < 0.05). The spray-drying process with higher energy plausibly induced molecular unfolding of FMP. As a consequence, several hydrophobic amino acid residues of protein molecules localized internally were more exposed [15]. The fluorescence probe ANS readily attaches to hydrophobic amino acids with an aromatic ring and has been employed to evaluate the S_o_ of proteins or protein complexes [15]. The S_o_ of FD-FMP and SD-FMP samples were reduced with the addition of TA, especially at higher concentrations (*p* < 0.05). The introduction of hydrophilic hydroxyl and carboxyl groups from TA could lower the hydrophobicity of proteins [12]. Zhang et al. [25] stated that oxidized phenolic compounds aided the protein molecules in enhancing a conformational change with greater hydrophilicity. ANS fluorescence intensity generally decreased in the hydrophilic environment [12]. The covalent interaction of proteins and oxidized phenolic compounds under alkaline conditions generally caused the blocking of hydrophilic groups, including thiol and charged amino groups. As a result, the S_o_ of FD-FMP and SD-FMP samples was decreased, particularly when the highest level of TA (0.5%) was used (*p* < 0.05). Similarly, Zhan, Li, Wang, Shi, Li, and Sheng [12] reported that S_o_ was decreased for a TA/sodium caseinate nano complex with increasing TA concentration. Chen, Jiang, Chen, Liu, and Kong [26] also documented that a significant decrease in S_o_ was obtained in porcine plasma protein hydrolysates modified by TA with increasing concentrations.

Total SH group contents (TSC) of FD-FMP and SD-FMP modified with TA at varying concentrations are presented in Table 1. The SH groups are relatively active groups in the proteins [27]. There was a gradual decrease in the TSC of both samples (FD-FMP and SD-FMP) with increasing TA concentrations (*p* < 0.05). TA is a water-soluble polyphenol with several hydroxyl groups that become oxidized to quinones in an alkaline condition in the presence of O_2_, which further reacted with amino or SH groups on the protein to form covalent bonds via Schiff base and Michael addition reactions [28]. Moreover, at high pH, side chain groups of cysteine and lysine covalently reacted to form C-S to C-N bonds, respectively. Therefore, the reduction in the SH group has been considered to be mediated by the covalent modification of proteins [29]. At 0.5% TA, the lowest TSC was observed in both samples, FD-FMP and SD-FMP (*p* < 0.05). Covalent interaction more likely facilitated interaction between TA and SH groups in FMP under alkaline conditions, thus reducing TSC. The results were in line with the S_o_ values of the FD-FMP and SD-FMP samples (Table 1). Similarly, Malik et al. [30] reported that TSC was lower upon the interaction of proteins with polyphenols.

On the other hand, disulfide bond content (DSBC) increased for both samples with increasing TA concentrations (Table 1). DSBC were augmented with the coincidental decrease in TSC of both samples. Molecular interaction between FMP and TA modified the protein structure and exposed SH groups, which were further oxidized to disulfide bonds (DSB) [28]. The sulfurs from two distinct cysteine amino acids can form a disulfide bridge through an oxidation process. As a result, DSBC was upsurged with increasing TA concentrations, particularly at 0.5%. Higher DSBC was observed in all SD-FMP samples than that of FD-FMP samples (*p* < 0.05). FMP plausibly unfolded or underwent conformational changes during the spray-drying, in which the cysteine residues were exposed. This made the SH groups more susceptible to oxidation into DSB. Therefore, the SH groups of SD-FMP were more prone to disulfide bond formation, which was aided by the structural changes during spray-drying. The results were in agreement with TSC, where the decline in the TSC of both samples could be attributed to the formation of DSB. The results suggested that interaction between FMP and TA with varying concentrations, followed by drying with different methods, could modify or alter the protein structure in different fashions.

### 3.2. Fluorescence Intensity

The fluorescence intensity of FD-FMP and SD-FMP modified with TA at varying concentrations is shown in Figure 1. The interactions between proteins and polyphenols have been extensively investigated by using fluorescence emission spectroscopy. Proteins have intrinsic emission fluorescence, primarily caused by tryptophan, tyrosine, and phenylalanine residues. Tryptophan has the highest quantum yield among all three amino acids. It is excited at wavelength ~280 nm and emits fluorescence, which appeared as a peak from 300 to 400 nm [31]. For the control sample of SD-FMP (without TA), higher fluorescence intensity was found compared to that of the FD-FMP sample. In general, the fluorescence emission spectral properties were altered when these aromatic residues were exposed to varying degrees. Spray-drying more likely induced the molecular unfolding of FMP. As a result, aromatic amino acids in FMP, such as tryptophan, were exposed during spray-drying, and FMP emits fluorescence under excitation. The results were in accordance with the S_o_, in which the SD-CON sample showed higher S_o_ values than the FD-CON sample. A decrease in fluorescence intensity of both FD-FMP and SD-FMP was noted after the addition of TA, particularly at a higher level (Figure 1). The lowered fluorescence intensity suggested that the interaction between FMP and TA plausibly led to energy transfer and remodeled the secondary structure of FMP, which quenched or masked fluorescent residues in FMP. Similar results were reported by Poklar Ulrih [31] regarding protein–polyphenol interactions. Chen, Jiang, Chen, Liu, and Kong [26] also documented that the fluorescence intensity of modified porcine plasma protein hydrolysates was significantly reduced with augmenting concentrations of TA. The results reconfirmed that the addition of TA followed by drying could alter the secondary structure of FMP to a certain extent, but the degree of changes depended on the drying methods used.

### 3.3. Fourier Transform Infrared (FTIR) Spectra

FTIR was used to elucidate the change in functional groups or protein structure. The FTIR spectra at wavenumbers ranging from 4000 to 400 cm^−1^ of FD-FMP and SD-FMP that are modified with varying TA concentrations are displayed in Figure 2. In general, all the FTIR spectra consisted of a broad and strong absorption band at 3276–3279 cm^−1^ (amide A), which is associated mainly with O–H stretching vibration [32]. The absorption band at 2925–2942 cm^−1^ (amide B) was caused by C–H stretching vibration. With the addition of TA, the amide A peak was shifted from 3270 cm^−1^ (FD-FMP-CON) to 3262 cm^−1^ (FD-FMP-0.5TA). Similarly, a peak was shifted from 3273 cm^−1^ (SD-FMP-CON) to 3265 cm^−1^ (SD-FMP-0.5TA). Broad bands in the range from 3600 to 3100 cm^−1^ were ascribed to the O–H stretching of the OH group or phenolic structure of TA [33]. TA has numerous reactive OH groups, which are able to form complexes with proteins. N–H and O–H groups are capable of forming hydrogen bonds or hydrophobic interactions within proteins or with polyphenols [34]. The shift of wavenumbers indicated that FMP was hydrogen bonded with TA and confirmed the formation of the FMP-TA complex. The changes in the secondary structure of proteins are typically related to the shifts of the amide I band (1700–1600 cm^−1^, C=O peptide bond) and the amide II band (1600–1500 cm^−1^, N–H bending and C–N stretching). Interaction with polyphenols can be monitored via the intensity changes and spectral shift of these two bands, which are linked with protein secondary structure [35]. With the addition of TA, amide I and amide II bands of FD-FMP and SD-FMP samples were shifted to different values. With the change in the positions of amide I and amide II bands, it can be inferred that FMP formed the complex with TA through C=O, C–N, and N–H.

Deconvolution analysis of bonds via the second derivation in the amide I region of FTIR spectra gives insight for the secondary structural components of the FMP (Table 2). The secondary structure contents of FD-FMP and SD-FMP were composed mainly of a β-sheet. The FD-FMP-CON sample had slightly higher β-sheet values than the SD-FMP-CON sample, indicating higher intermolecular hydrogen bonds of the former. With the addition of TA at different concentrations, the contents of the β-sheet, β-turn, and random coil were changed differently. With augmenting TA concentrations, the β-sheet of FD-FMP increased, while the β-sheet of SD-FMP remained constant. Both FD-FMP and SD-FMP had a slight decrease in β-turn as TA concentrations increased. A similar random coil of FD-FMP was observed, regardless of TA concentrations. However, the decrease in the random coil was attained in the presence of TA at all levels. Hasni et al. [36] reported that the conformation of proteins could be changed by polyphenol interactions, mainly due to conversion between β-sheet, β-turn, and random coil. The results suggested that interactions of FMP and TA caused changes in the secondary structure of FMP, which could lead to an improvement in the functionalities of the dried FMP-TA complex.

### 3.4. Protein Patterns

SDS-PAGE patterns of FD-FMP and SD-FMP modified with TA at varying concentrations under non-reducing and reducing conditions are displayed in Figure 3. The control sample of FD-FMP and SD-FMP had similar distinctive protein bands of tropomyosin (TM, 35 kDa), actin (AC, 45 kDa), and myosin heavy chain (MHC, 200 kDa) (Figure 3, Lane C). Differences in protein patterns were observed under non-reducing and reducing conditions for both FD-FMP and SD-FMP samples. SDS and β-mercaptoethanol were added in the SDS-PAGE experiment to break down the non-covalent interactions in proteins. The results indicated that FMP was stabilized by several disulfide bonds. Overall, protein band intensity was decreased with the increasing concentration of TA when compared to the control sample. Moreover, a high molecular weight polymer was found on the stacking gel in all samples containing TA. Covalent modification of proteins by phenolic oxidation products generated at alkaline pH was reported by Prodpran et al. [37]. Therefore, covalent interactions contributed to cross-linking between FMP and TA, in which the molecular weight of the complex increased. Lower protein band intensity was observed in SD-FMP samples than in FD-FMP samples. The results were in agreement with DSBC, in which higher DSBC was noted in SD-FMP samples (Table 1). Therefore, the SDS-PAGE protein pattern could also be used to confirm the covalent interactions between FMP and TA. Moreover, drying methods and TA concentrations could covalently modify the structure of FMP differently, which could improve emulsifying properties to varying degrees.

### 3.5. Emulsifying Properties

Emulsifying activity index (EAI) and emulsion stability index (ESI) of FD-FMP and SD-FMP modified with TA at varying concentrations are displayed in Figure 4. EAI corresponds to the ability of proteins to localize surrounding oil droplets, and it is measured as the unit area of the interface covered per unit weight of protein [38]. The EAI of the SD-FMP-CON sample was higher than the FD-FMP-CON sample (*p* < 0.05), indicating the higher capability of the SD-FMP to migrate to the interface and stabilize the oil droplets. This ability of proteins to occupy the interface is generally governed by the augmented solubility and surface hydrophobicity of proteins [39]. The presence of more hydrophobic domains in the proteins resulted in enhanced interactions between the proteins and oil [40]. Compared to the FD-FMP-CON sample, SD-FMP-CON samples had higher hydrophobic areas, as revealed by a higher S_o_ (Table 1). The EAI of FD-FMP and SD-FMP samples was lowered with the addition of TA. The results suggested that the decreased EAI was mostly ascribed to the lower S_o_ of FMP after the addition of TA (Table 1). Similarly, Zhan, Li, Wang, Shi, Li, and Sheng [12] reported that the diffusion coefficient of TA/sodium caseinate nano complexes was lowered with increasing TA concentration. In general, proteins or protein complex adsorption at the interface involves diffusion, penetration, structural deformation, and rearrangement, wherein a multilayer can be formed [41]. Hence, the diffusion of FMP-TA complexes to the water/oil interface gives rise to a change in the adsorption kinetics. The FMP-TA complex was unable to rapidly migrate at the interface and cover oil droplets efficiently. This was plausibly due to a greater viscosity of the FMP-TA complex solution, thus decreasing its migration to the interface. As a result, the EAI was lowered in both samples (FD-FMP and SD-FMP) when modified by TA.

On the other hand, the ESI values of FD-FMP and SD-FMP samples were augmented with increasing TA concentration. Generally, high ESI values of samples were attributed to a thick film at the interface formed by protein or protein complex [42]. With the addition of polyphenols, bridging between proteins and proteins favored intermolecular interactions and molecular rearrangement at the interface, thus allowing the formation of a thick interfacial layer [12]. Bandyopadhyay et al. [43] reported that when the protein/epigallocatechin gallate (EGCG) ratio increased, EGCG bridged the soluble particles, leading to metastable colloids. The results suggested that FMP-TA complexes were more potent in forming the cross-linked network, in which oil droplets were entrapped effectively. A thick interfacial layer formed by a cross-linked network could prevent the coalescence or flocculation of oil droplets, resulting in a more stable emulsion. Similarly, Chen, Jiang, Chen, Liu, and Kong [26] documented that the EAI of modified porcine plasma protein hydrolysates significantly decreased, but the ESI was increased when TA concentration upsurged. Overall, FD/SD FMP modified with higher TA concentration could reduce the EAI, but it conversely improved ESI. Therefore, the spray-dried FMP-TA complex modified with 5% TA could be advantageous for use in food emulsion.

### 3.6. Characteristics and Storage Stability of Pickering Emulsion (PE) Stabilized by FMP-TA Complex

#### 3.6.1. Confocal Laser Scanning Microscopic (CLSM) Images

Microscopic images of oil droplets in PE prepared by FD-FMP-0.5TA and SD-FMP-0.5TA as emulsifiers/stabilizers visualized by CLSM are illustrated in Figure 5. For both PE samples, the oil droplets were more homogeneous in size and shape, indicating monodispersed emulsions. At day 0, the oil droplets in the SD-FMP-0.5TA emulsion were slightly lower in size when compared to those found in the FD-FMP-0.5TA emulsion. The results indicated that the former was more efficient than the latter in reducing oil droplets during emulsification. The FMP-TA complex augmented the entrapment of smaller oil droplets in the SD-FMP-0.5TA emulsion more effectively than the other. However, after storage for 45 days, the coalescence or collapse was more evident in the FD-FMP-0.5TA emulsion than in the SD-FMP-0.5TA emulsion (Figure 5). Because of the unfavorable interaction between water and oil, emulsions are thermodynamically unstable [44]. Consequently, its physical properties were more likely changed during prolonged storage via several mechanisms, mainly via flocculation and coalescence. Therefore, droplets coalesced to form larger-sized droplets. Thus, the SD-FMP-0.5TA complex was more effective as an emulsifier/stabilizer in oil-in-water emulsion than the FD-FMP-0.5TA sample and also yielded a more stable emulsion.

#### 3.6.2. Droplet Size

The droplet size of both emulsions expressed as *d*_32_ and *d*_43_ are shown in Table 3. At day 0, the SD-FMP-0.5TA emulsion sample had a smaller *d*_32_ (1.49 ± 0.07 µm) than the other (1.63 ± 0.03 µm). Nevertheless, the *d*_32_ values of the SD-FMP-0.5TA and the FD-FMP-0.5TA emulsions were increased to 1.73 ± 0.03 µm and 1.77 ± 0.03 µm after 45 days of storage time, respectively. The *d*_32_ is related to the average surface area of droplets exposed to the continuous phase per unit volume of the emulsion. A smaller *d*_32_ means a higher specific surface area [38]. A similar trend was also found for *d*_43_, in which the SD-FMP-0.5TA emulsion showed a smaller *d*_43_ (3.25 ± 0.15 µm) than the FD-FMP-0.5TA emulsion (3.46 ± 0.11 µm) at day 0, which was increased to 4.05 ± 0.15 µm and 4.50 ± 0.89 µm after 45 days of storage, respectively. The *d*_43_ is the sum of the volume ratio of droplets in each size class multiplied by the mid-point diameter of the size class. At day 0, the SD-FMP-0.5TA emulsion had a smaller particle than the FD-FMP-0.5TA emulsion. The former had higher hydrophobic areas than the latter, as revealed by the augmented S_o_ (Table 1). The presence of more hydrophobic areas in the FMP-TA complex obtained after spray-drying resulted in enhanced interactions and molecular rearrangement at the interfacial layer of oil and water. As a result, SD-FMP modified with 0.5% TA could stabilize an emulsion with smaller oil droplets. However, emulsion stability was slightly decreased after 45 days of storage. Molecular rearrangements at the oil/water interface plausibly weakened the cross-linking of the network formed by the FMP-TA complex. The increases in both *d*_32_ and *d*_43_ (Table 3) revealed that the oil droplets coalesced. The formation of larger oil droplets of the FD-FMP-0.5TA emulsion indicated poor emulsion stability. The variations in the size of oil droplets were also confirmed via CLSM (Figure 4), in which the FD-FMP-0.5TA emulsion had the slightly larger oil droplets at 45 days. Therefore, FMP modified with 5% TA followed by spray-drying was effective as an emulsifier/stabilizer in an oil-in-water emulsion.

#### 3.6.3. Coalescence Index (*C_i_*) and Flocculation Factor (*F_f_*)

The *C_i_* and *F_f_* of the FD-FMP-0.5TA- and the SD-FMP-0.5TA-stabilized emulsions are tabulated in Table 3. After 45 days, a higher *C_i_* was noted in the former (29.80 ± 2.31) than in the latter (24.61 ± 1.13), suggesting that more coalescence occurred in the former. In contrast, the *F_f_* of both emulsion samples was reduced, indicating that individual oil droplets merged to form the larger oil droplets. Consequently, the floc was diminished. The SD-FMP-0.5TA emulsion with a lower *C_i_* had a lower rate of changes in *d*_32_ and *d*_43_ compared to the FD-FMP-0.5TA emulsion (Table 3). The emulsion stability depends mainly on the protective barrier film formed around oil droplets that prevents coalescence [20]. The stability of emulsions resulted mainly from the strong cross-linked network of the FMP-TA complex that acts as a protective barrier against coalescence. However, the FMP-TA complex adsorbed at the interface plausibly led to some alteration or rearrangements at the interface, leading to the aggregation of droplets and enhanced coalescence. Overall, FMP modified with 5% TA followed by spray-drying was advantageous for the preparation of food-grade, particle-based PE.

#### 3.6.4. Rheological Properties

Storage modulus (G′) and loss modulus (G″) of emulsions using FD-FMP-0.5TA and SD-FMP-0.5TA as stabilizers/emulsifiers at day 0 and day 45 are displayed in Figure 6, respectively. In general, G′ was lower than G″, irrespective of the FMP-TA complex used. This suggested that emulsions had more viscous behavior than elastic behavior [45]. Both emulsions had higher G′ and G″ at day 0 than at day 45. This suggested that more stress was needed for the emulsion to flow because of the higher interaction of droplets and compact structures at day 0 [22]. However, the G′ and G″ values were decreased at day 45, suggesting the reduced interactions of droplets or some polymers in both emulsions. This was mostly associated with coalescence, where the larger droplets had reduced viscosity [38]. These results were in line with the larger size of droplets in the emulsions at day 45 (Table 3, Figure 5). The lowered modulus values might be associated with structural rearrangements of oil droplets with extended storage [22]. The SD-FMP-0.5TA stabilized emulsion showed a higher modulus (G′ and G″) than the FD-FMP-0.5TA emulsion before and after storage. A higher interaction of the FMP-TA complex more likely formed a highly viscous network at the interface. The interfacial film thickness and the surface load of the emulsifiers had a profound impact on the stability of the emulsion [46]. The superior viscoelastic property was found in the SD-FMP-0.5TA emulsion, indicating a thick interfacial layer of the FMP-TA complex with greater viscosity and elasticity.

The viscosity of both emulsions stored at day 0 and 45 as a function of shear rate is shown in Figure 7a. Viscosity generally dropped as the shear rate increased, indicating shear thinning behavior [38]. For both samples, higher viscosity was observed on day 0. Nevertheless, viscosity decreased after 45 days of storage. The number of droplets per unit volume dropped, as indicated by the increased droplet size. As a result, more mobility or less resistance to flow was obtained, as shown by the decreased viscosity [22]. At a low shear rate, the SD-FMP-0.5TA emulsion showed slightly higher viscosity than the FD-FMP-0.5TA emulsion. This was ascribed to the effective cross-linked network formed by the SD-FMP-TA complex. Nevertheless, a lower reduction in viscosity was noted when the shear rate increased in the SD-FMP-0.5TA emulsion than the other. The stability of oil droplets during emulsification and prolonged storage was facilitated by the viscoelastic interfacial layer [47]. The SD-FMP-0.5TA sample was more surface active and possessed greater molecular flexibility than the FD-FMP-0.5TA sample, which was revealed by the higher EAI of the former (Figure 4). Additionally, the higher hydrophobic region in the SD-FMP-TA complex could interact more efficiently at the interface, leading to the formation of the compact and thick interfacial layer. As a result, saturated coverage on the surface of droplets was developed. The result was in line with the higher ESI (Figure 4). As a result, lower coalescence and flocculation of the droplets were found, as indicated by the lower *C_i_* and *F_f_* in the SD-FMP-0.5TA emulsion compared to FD-FMP-0.5TA emulsion (Table 3).

The flow curves of both emulsions at day 0 and 45 are displayed in Figure 7b. The flow curves of the shear-thinning emulsions were linear on a log–log plot of shear stress versus shear rate, indicating pseudoplastic behavior. Hence, a power-law model could be fitted to each curve:$$\tau =k\gamma^{n}$$
where τ is the shear stress (Pa), γ is the shear rate (s^−1^), *k* is the fluid consistency index (Pa·s^n^), and n is the flow behavior index. Generally, shear stress increased with augmenting shear rate, representing a non-Newtonian behavior [38]. The emulsions at day 45 had lower shear stress compared to day 0. This led to a decrease in flow behavior after storage. Viscosity represents the ratio of shear stress to shear rate [38]. Shear stress gradually increased with the coincidental decrease in viscosity [22]. Overall, the SD-FMP-0.5TA emulsion showed greater viscosity at a high shear rate than the FD-FMP-0.5TA emulsion at both storage times.

## 4. Conclusions

The interaction between FMP and TA at varying concentrations that was followed by spray-drying or freeze-drying directly caused structural modifications, as confirmed by the reduced surface hydrophobicity and total sulfhydryl contents compared to the control. TA was covalently linked to FMP via covalent bonds, including disulfide bonds. The formation of the FMP-TA complex was also confirmed by FTIR spectra, fluorescence, and protein patterns. The spray-dried FMP-TA complex showed greater structural changes than the freeze-dried FMP-TA complex. The emulsion stabilized by the SD-FMP-0.5TA complex rendered higher ESI but lowered EAI. Moreover, the rheological properties of the SD-FMP-0.5TA emulsion were more maintained than the FD-FMP-0.5TA emulsion, after the storage of 45 days. Therefore, the spray-dried FMP-TA complex modified with 0.5% TA could be used as a potential food-grade stabilizer/emulsifier for Pickering emulsions with enhanced emulsifying properties.

## Figures and Tables

**Figure 1 foods-12-01556-f001:**
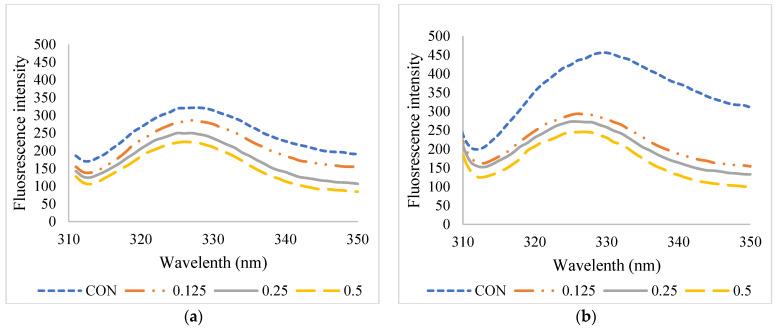
Fluorescence intensity of (**a**) freeze-dried (FD) and (**b**) spray-dried (SD) fish myofibrillar proteins (FMP) modified with tannic acid (TA) at varying concentrations. CON: No TA added. Numbers represent the level of tannic acid (%) used for modification.

**Figure 2 foods-12-01556-f002:**
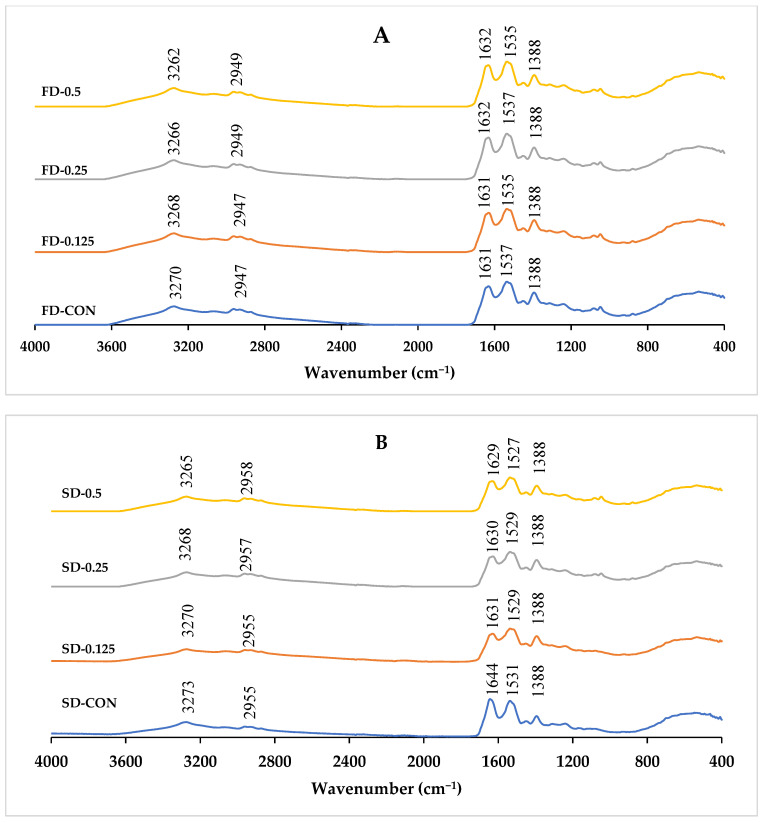
FTIR of (**A**): freeze-dried (FD), and (**B**): spray-dried (SD) fish myofibrillar proteins (FMP) modified with tannic acid (TA) at varying concentrations. CON: No TA added. Numbers represent the level of tannic acid (%) used for modification.

**Figure 3 foods-12-01556-f003:**
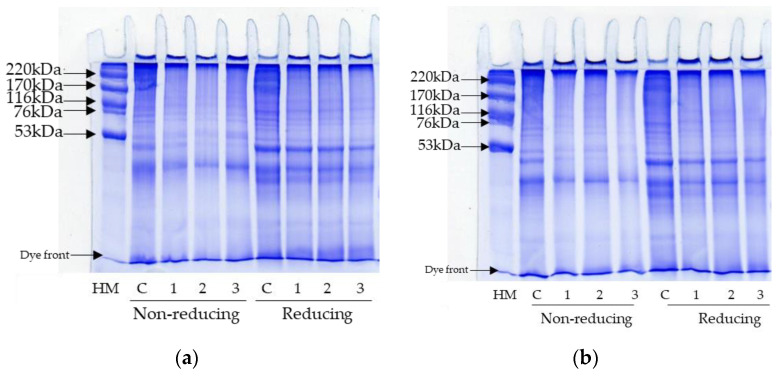
Protein patterns of (**a**) freeze-dried (FD) and (**b**) spray-dried (SD) fish myofibrillar proteins (FMP) modified with tannic acid (TA) at varying concentrations under non-reducing and reducing conditions. C: control sample (without TA), Number (1, 2, 3) denotes TA concentration of 0.125%, 0.25%, and 0.5%, respectively. HM: High molecular weight marker.

**Figure 4 foods-12-01556-f004:**
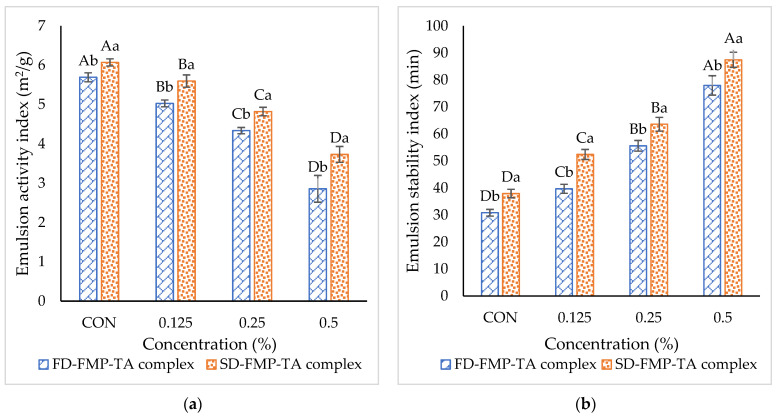
(**a**) Emulsion activity index (EAI) and (**b**) emulsion stability index (ESI) of emulsions stabilized by freeze-dried (FD) and spray-dried (SD) fish myofibrillar proteins (FMP) modified with tannic acid (TA) at varying concentrations. Different uppercase letters on the bars within the same sample denote significant differences (*p* < 0.05). Different lowercase letters on the bars within the same concentration of the FMP-TA complex denote significant differences (*p* < 0.05). Error bars represent standard deviation (n = 3).

**Figure 5 foods-12-01556-f005:**
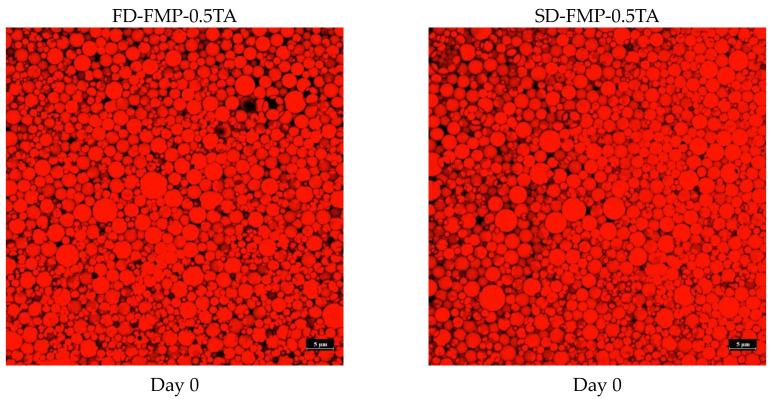
Confocal laser scanning microscopic (CLSM) images of emulsions prepared by freeze-dried (FD) and spray-dried (SD) fish myofibrillar proteins (FMP) modified with 0.5% tannic acid (TA) and stored at day 0 and day 45. Ellipsoidal marks indicate the coalescence of oil droplets.

**Figure 6 foods-12-01556-f006:**
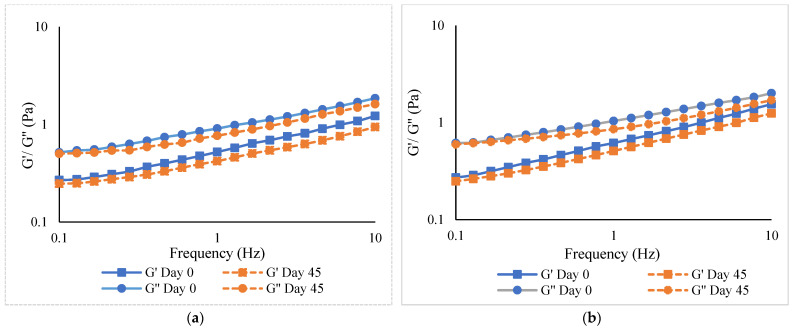
Storage modulus—G′ and loss modulus—G″ of emulsions stabilized by (**a**) freeze-dried (FD) and (**b**) spray-dried (SD) fish myofibrillar protein (FMP) modified with 0.5% tannic acid at day 0 and 45 of storage at 28 °C.

**Figure 7 foods-12-01556-f007:**
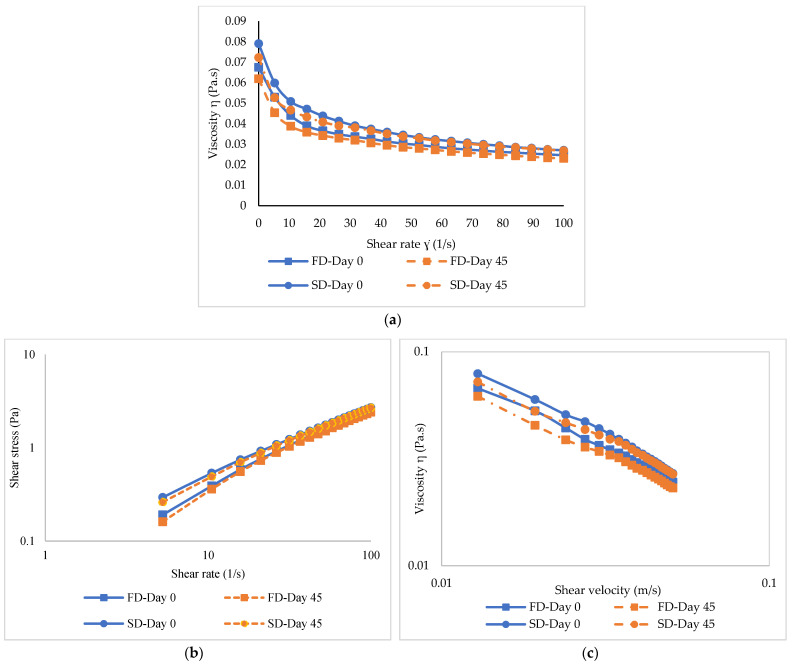
Viscosity (**a**) and flow curves (**b**,**c**) of emulsions stabilized by freeze-dried (FD) and spray-dried (SD) fish myofibrillar protein (FMP) modified with 0.5% tannic acid at day 0 and 45 of storage at 28 °C.

**Table 1 foods-12-01556-t001:** Surface hydrophobicity, total sulfhydryl, and disulfide bond contents of freeze-dried (FD) and spray-dried (SD) fish myofibrillar proteins (FMP) modified with tannic acid (TA) at varying concentrations.

Samples (TA, %)	Surface Hydrophobicity (S_o_) (×10^4^)		Total Sulfhydryl Content (TSC) (μmol/g Protein)		Total Disulphide Bond Content (DSBC) (μmol/g Protein)	
	FD	SD	FD	SD	FD	SD
CON	15.72 ± 0.30 Ab	19.22 ± 0.23 Aa	2.00 ± 0.05 Ab	2.68 ± 0.08 Aa	6.42 ± 0.29 Ab	7.72 ± 0.36 Aa
0.125	12.49 ± 0.23 Bb	18.42 ± 0.24 Ba	1.57 ± 0.15 Bb	2.42 ± 0.10 Ba	7.15 ± 0.29 Bb	8.38 ± 0.32 Ba
0.25	8.50 ± 0.44 Cb	12.63 ± 0.32 Ca	1.35 ± 0.03 Cb	2.22 ± 0.08 Ca	7.79 ± 0.19 Cb	9.26 ± 0.22 Ca
0.5	2.27 ± 0.27 Db	7.63 ± 0.22 Da	0.72 ± 0.10 Db	1.97 ± 0.11 Da	8.33 ± 0.11 Db	10.41 ± 0.22 Da

Values are mean ± standard deviation (*n* = 3). Different uppercase letters in the same column indicate a significant difference (*p* < 0.05). Different lowercase letters in the same row within the same parameter indicate a significant difference (*p* < 0.05). TA: Tannic acid. CON: No TA added.

**Table 2 foods-12-01556-t002:** Secondary structure of freeze-dried (FD) and spray-dried (SD) fish myofibrillar proteins (FMP) modified with tannic acid (TA) at varying concentrations.

Samples (TA, %)	β-Sheets (%)		β-Turn (%)		Random Coil (%)	
	FD	SD	FD	SD	FD	SD
CON	34.87	34.33	22.78	24.67	16.62	17.96
0.125	35.15	34.16	22.61	22.71	16.25	16.25
0.25	42.30	34.48	20.96	22.90	16.56	16.56
0.5	40.18	34.21	19.88	23.20	16.63	16.63

TA: Tannic acid. CON: No TA added.

**Table 3 foods-12-01556-t003:** Droplet size and stability of oil-in-water emulsion stabilized by freeze-dried (FD) and spray-dried (SD) fish myofibrillar proteins (FMP) modified with 0.5% tannic acid.

Samples	Storage Time (Days)	*d*_32_ (µm)	*d*_43_ (µm)	*F_f_*	*C_i_*
FD-FMP-0.5TA	0	1.63 ± 0.03 A	3.46 ± 0.11 A	0.94 ± 0.02 A	-
	45	1.77 ± 0.03 a	4.50 ± 0.89 a	0.89 ± 0.03 a	29.80 ± 2.31 a
SD-FMP-0.5TA	0	1.49 ± 0.07 B	3.25 ± 0.15 B	0.95 ± 0.01 A	-
	45	1.73 ± 0.03 b	4.05 ± 0.15 b	0.87 ± 0.00 b	24.61 ± 1.13 b

Values are mean ± standard deviation (n = 3). Different uppercase letters in the same column within the same storage time (day 0) indicate a significant difference (*p* < 0.05). Different lowercase letters in the same column within the same storage time (day 45) indicate a significant difference (*p* < 0.05).

## Data Availability

The data presented in this study are available in the article.

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
