# Peer review of "Pickering Emulsion Stabilized by Fish Myofibrillar Proteins Modified with Tannic Acid, as Influenced by Different Drying Methods"

_foods, 2023, doi:10.3390/foods12071556_

Round 1
Reviewer 1 Report
In this paper, the author reported a novel Pickering emulsion that was stabilized by TA-modified FMP. The characteristic of FMP was first tested. Subsequently, TA-modified FMP was synthesized and applied as a stabilizer for Pickering emulsion. But there are some deficiencies that need to be improved.
1. There are several reports before about the same concept as this. The novelty needs to be highlighted, e.g. Prodpran, T., Benjakul, S., & Phatcharat, S. (2012). Effect of phenolic compounds on protein cross-linking and properties of film from fish myofibrillar protein. International journal of biological macromolecules, 51(5), 774-782.
Balange, A. K., & Benjakul, S. (2010). Cross-linking activity of oxidised tannic acid towards mackerel muscle proteins as affected by protein types and setting temperatures. Food Chemistry, 120(1), 268-277.
Zhong, Q., Li, H., Deng, S., Ren, Y., Kong, B., & Xia, X. (2021). Tannic acid-induced changes in water distribution and protein structural properties of bacon during the curing process. LWT, 137, 110381.
2. Missing zeta-potential result of FMP-TA
3. Missing Pickering emulsion stability data.
4. The visualization of the paper needs to be improved.
Author Response
Responses to reviewer
Reviewer: 1
In this paper, the author reported a novel Pickering emulsion that was stabilized by TA-modified FMP. The characteristic of FMP was first tested. Subsequently, TA-modified FMP was synthesized and applied as a stabilizer for Pickering emulsion. But there are some deficiencies that need to be improved.
***** We are thankful to the reviewer for understanding our work. We have considered the suggestion made by the reviewer and have accordingly corrected in the revised text.
- There are several reports before about the same concept as this. The novelty needs to be highlighted, e.g. Prodpran, T., Benjakul, S., & Phatcharat, S. (2012). Effect of phenolic compounds on protein cross-linking and properties of film from fish myofibrillar protein. International journal of biological macromolecules, 51(5), 774-782.
Balange, A. K., & Benjakul, S. (2010). Cross-linking activity of oxidised tannic acid towards mackerel muscle proteins as affected by protein types and setting temperatures. Food Chemistry, 120(1), 268-277.
Zhong, Q., Li, H., Deng, S., Ren, Y., Kong, B., & Xia, X. (2021). Tannic acid-induced changes in water distribution and protein structural properties of bacon during the curing process. LWT, 137, 110381.
***** Thank you for the suggestion. The use of protein/polyphenol complexes has been widely reported and studied. However, the objectives in aforementioned research articles were totally different among different researches as explained below:
1) For the first paper, protein/polyphenol interactions were used for improvement of fish myofibrillar protein film properties.
2) In the second paper, oxidized tannic acid was used as protein cross-linker, which could strengthen the gel from mackerel muscle protein.
3) For the third paper, tannic acid was used to improve physicochemical properties of cured bacon.
In our study, we used yellow stripe trevally (Selaroides leptolepis), which is a common dark-fleshed fish found in Southern Thailand. This is underutilized species generally known as a low-value marine resource with minimal market value and it is very susceptible to spoilage. Therefore, the use of fish myofibrillar protein (FMP) modified with tannic acid (TA) in form of solid particles, which could be further used for preparation of Pickering emulsion (PE), offers a novel alternative natural emulsifier/stabilizer and also increased its market value. We provided the new information on a development of novel food-grade particles-based PE from marine source with high nutritive value. Moreover, the long-term emulsion stability of PE was achieved by the adsorption of FMP-TA complexed solid particles on the oil droplet surface and the development of strong layer around oil droplets.
The more information regarding the novelty of research has been added in revised manuscript. Please see line number 63-66.
- Missing zeta-potential result of FMP-TA
***** Thank you for the comment. Sorry. We did not determine zeta-potential in this study. It is well known that in a typical oil-in-water emulsion, the more negative the zeta potential is, the more stable the emulsion is obtained. For emulsion stability measurement, we determined emulsifying activity index (EAI) and emulsion stability index (ESI), which is a simple and inexpensive method for examining the stability of an emulsion. Several researchers also did not report the zeta-potential for the emulsion and only reported EAI and ESI. For example,
Ashaolu, T. J., & Zhao, G. (2020). Fabricating a Pickering stabilizer from okara dietary fibre particulates by conjugating with soy protein isolate via Maillard reaction. Foods, 9(2), 143.
Xie, H., Wei, X., Liu, X., Bai, W., & Zeng, X. (2023). Effect of polyphenolic structure and mass ratio on the emulsifying performance and stability of emulsions stabilized by polyphenol-corn amylose complexes. Ultrasonics Sonochemistry, 106367.
In the present study, several parameters related with the stability of emulsion have been examined to indicate the potential of fish protein-tannic acid complex to stabilize Pickering emulsion. However, the valuable point raised by the reviewer has been taken into our consideration. We will measure zeta potential in our on-going research as suggested by the reviewer. Thank you so much.
- Missing Pickering emulsion stability data.
***** Thank you for the comment. Characteristics and storage stability of Pickering emulsion were evaluated at day 0 and day 45 by confocal laser scanning microscopic (CLSM) images, particle size, coalescence index, flocculation factor, rheological properties, and flow behavior. Please see section 3.6.
- The visualization of the paper needs to be improved.
***** Thank you for the comment. We have enhanced figure resolution for better visualization. Additionally, graphical abstract was also prepared for the visualization of research data.

Reviewer 2 Report
The article, although the topic it deals with is interesting, requires a thorough review in some aspects, especially in rheological characterization. Some dubious claims are made and even some self-quoted ones that can be misinterpreted by readers.
I do not agree with the definition of emulsion. They are thermodynamically unstable systems, although they can be kinetically stable. In addition, the amount of emulsifier does not necessarily have to be small. I strongly suggest using a definition of a book that fits what an emulsion really is.
There are other recent articles on the use of food stabilizers (such as zein) for the development of PE. Please expand the bibliography/number of references.
Line 124. Please specify the type of homogenizer used
Since we are talking about drops and not particles, I think it is appropriate to replace particle size with droplet size
Why were polydispersity or span data not included in the droplet size analysis?
The use of the volumetric diameter to study destabilization mechanisms such as coalescence or flocculation has already been used in numerous studies. Some of these studies signed by the way, and without obviously mentioning studies or names, by this humble reviewer. This implies that the statement, which incidentally does not exactly conform to reality, carried out in line 412 and self-cited with reference 16 seems to me excessively risky.
Droplet size distributions should be added, if possible over time, not only to see the polydispersity but also to be able to differentiate between coalescence and Ostwald ripening.
The title of section 3.6.4 specifies that a rheological property has been determined, when I understand that it refers to viscoelastic properties or oscillatory shear tests, because flow properties are also rheological properties. Frequency scans have been carried out without specifying whether stress sweeps (or at what frequencies) have been carried out to determine the linear viscoelastic range.
A deeper analysis of frequency sweeps should be carried out (G´>G´´? gel-like behaviour?)
I suggest representing G ́ and G ́ ́ with the same color and different symbol
Figures 7A and 7B are actually the same considering the direct relationship between viscosity and stress. In addition, to better see the differences, viscosity versus shear velocity should be represented on a logarithmic scale.
The analysis of flow curves, which have not even been fitted, must be deeper.
Author Response
Response to reviewer
Reviewer 2
The article, although the topic it deals with is interesting, requires a thorough review in some aspects, especially in rheological characterization. Some dubious claims are made and even some self-quoted ones that can be misinterpreted by readers.
****Thank you so much for the time spent on our manuscript for quality and clarity improvement. All queries have been responded and the corrections have been made appearing as track changes in the manuscript.
I do not agree with the definition of emulsion. They are thermodynamically unstable systems, although they can be kinetically stable. In addition, the amount of emulsifier does not necessarily have to be small. I strongly suggest using a definition of a book that fits what an emulsion really is.
****The definition of emulsion has been revised from the well-known book. Authors do agree that small amount of emulsifier is not always required for emulsion. Such a phrase has been removed to avoid confusion. Please see line 43-44.
There are other recent articles on the use of food stabilizers (such as zein) for the development of PE. Please expand the bibliography/number of references.
****Thank you for the suggestion. The number of references regarding the development of PE using varying food stabilizers have been added in the revised text, especially in introduction (line 52-58).
Line 124. Please specify the type of homogenizer used.
****Thank you for the comment. The information regarding homogenizer has been added in revised manuscript. Please see line number 136.
Since we are talking about drops and not particles, I think it is appropriate to replace particle size with droplet size
****Authors agree with the reviewer. Therefore, the term ‘particle size’ has been replaced by ‘droplet size’ throughout the manuscript.
Why were polydispersity or span data not included in the droplet size analysis?
*****We use the machine ANALYSETTE 22 NanoTec, FRITSCH, Germany, in which span data were not reported. Authors did not use zeta potential analyzer in the present study. Sorry for the missing aforementioned information. However, the comment will be taken into consideration for our future work.
The use of the volumetric diameter to study destabilization mechanisms such as coalescence or flocculation has already been used in numerous studies. Some of these studies signed by the way, and without obviously mentioning studies or names, by this humble reviewer. This implies that the statement, which incidentally does not exactly conform to reality, carried out in line 412 and self-cited with reference 16 seems to me excessively risky.
*****Authors have removed the statement to avoid such weak or misleading statements. Moreover, some self-cited references have been replaced with the related references.
Droplet size distributions should be added, if possible over time, not only to see the polydispersity but also to be able to differentiate between coalescence and Ostwald ripening.
*****Authors would like to apologize for this. As mentioned previously, due to the limitation of instrument, only average size was reported. Regarding the stability, authors evaluated the Pickering emulsion at day 0 and day 45 by confocal laser scanning microscopic (CLSM) images, particle size, coalescence index, flocculation factor, and rheological properties. For the invaluable suggestion, we try to find out the appropriate instrument to obtain such scientific data for better clarity of our work in the future. Differentiation between coalescence and Ostwald ripening are very interesting and informative. Authors will extend the study on instability of emulsion in the near future following the reviewer’s suggestion.
The title of section 3.6.4 specifies that a rheological property has been determined, when I understand that it refers to viscoelastic properties or oscillatory shear tests, because flow properties are also rheological properties. Frequency scans have been carried out without specifying whether stress sweeps (or at what frequencies) have been carried out to determine the linear viscoelastic range.
******Authors agree to remove the title of section 3.6.5. ‘Flow behavior’ and merge under section 3.6.4. ‘Rheological properties’. Authors would like to apologize for the missing information. Frequency scans were carried out after stress sweep was performed to ensure that such a condition provide the linear range. The details have been provided in the methodology. Please see line 180-182.
A deeper analysis of frequency sweeps should be carried out (G´>G´´? gel-like behaviour?)
******Based on general knowledge from the text or peer-review papers, a higher value for the storage modulus (G′) than the loss modulus (G") generally indicates its gel/semi-solid behavior. However, when G′ is lower than G", it is suggested that the emulsion has more viscous behavior than elastic behavior. Sharma and Sangwai (2015) reported that nanoparticle–surfactant–polymer–salt (NSPS) stabilized Pickering emulsion showed lower G′ than G" in the tested frequency ranges, indicating a viscous-like behavior of NSPS emulsion system. Similarly, in the present study, we also found that lower G′ than G" at day 0 and both values were slightly lower at day 45.
Sharma, T., & Sangwai, J. S. (2015). Effects of electrolytes on the stability and dynamic rheological properties of an oil-in-water pickering emulsion stabilized by a nanoparticle–surfactant–polymer system. Industrial & Engineering Chemistry Research, 54(21), 5842-5852.
Zou, Y., van Baalen, C., Yang, X., & Scholten, E. (2018). Tuning hydrophobicity of zein nanoparticles to control rheological behavior of Pickering emulsions. Food Hydrocolloids, 80, 130-140.
Fuhrmann, P. L., Breunig, S., Sala, G., Sagis, L., Stieger, M., & Scholten, E. (2022). Rheological behaviour of attractive emulsions differing in droplet-droplet interaction strength. Journal of Colloid and Interface Science, 607, 389-400.
I suggest representing G ́ and G ́ ́ with the same color and different symbol
****Thank you so much. The changes have been done, following the reviewer’s suggestion. Please see Figure 6. The similar changes have been implemented for Figure 7 for better understanding.
Figures 7A and 7B are actually the same considering the direct relationship between viscosity and stress. In addition, to better see the differences, viscosity versus shear velocity should be represented on a logarithmic scale.
****The graphs have been plotted on logarithmic scale as per the reviewer’s suggestion.
The analysis of flow curves, which have not even been fitted, must be deeper.
****Thank you for the suggestion. Flow curves has been fitted and some discussion has been extended. Please see line number 511-517.
****All the insightful comments from the reviewer are highly appreciated. Those will be well taken into consideration for our future work. Authors would like to express our sincere thanks for the eye-opening comments/suggestion from the reviewer.

Reviewer 3 Report
The manuscript foods-2281568 entitled “Pickering emulsion stabilized by fish myofibrillar proteins modified with tannic acid as influenced by different drying methods" is a very interesting paper that addresses a current topic of high interest.
This is a great research work. It is well written; the introduction section does explain what was done on the topic analyzed. In the same way, a deep review of pertinent literature was used. The materials and methods are very clear allowing their reproduction in any laboratory. The scientific quality of the results and discussion section is so good. The results are presented in a clear and concise way. The authors discuss and compare their findings with the previous report on this field.
The only serious issue to improve will be:
The statistical assay must be improved. The statistical analysis established should be a two-factor ANOVA (TA concentration and dry method). Explain, please
In its current state, the level of English throughout your manuscript does not meet the journal's desired standard. There are some grammatical and spelling errors and full stops missing as well as instances of badly worded/constructed sentences. Please check the manuscript and refine the language carefully. I suggest that you should ask several colleagues who are skilled authors to check the English before your submission.
The main content of the abstract should include the brief purpose of the research, the principal result, and the major conclusion. The abstract, in the present form, is not adequate. Additionally, (i) the authors must state the revised justification in the abstract to support the study; (ii) some results should be nice.
I have not found a sign that the appropriate statistical analysis has been done. When we say that a compound is greater than another, we must be supported by the statistical analysis, indicating the p-value (p <0.05 or p> 0.05)
Author Response
Responses to reviewer
Reviewer: 3
The manuscript foods-2281568 entitled “Pickering emulsion stabilized by fish myofibrillar proteins modified with tannic acid as influenced by different drying methods" is a very interesting paper that addresses a current topic of high interest.
This is a great research work. It is well written; the introduction section does explain what was done on the topic analyzed. In the same way, a deep review of pertinent literature was used. The materials and methods are very clear allowing their reproduction in any laboratory. The scientific quality of the results and discussion section is so good. The results are presented in a clear and concise way. The authors discuss and compare their findings with the previous report on this field.
***** We are thankful to the reviewer for the encouraging and positive comments to improve the manuscript. All queries have been responded and the corrections have been made in the revised manuscript.
The only serious issue to improve will be:
The statistical assay must be improved. The statistical analysis established should be a two-factor ANOVA (TA concentration and dry method). Explain, please.
***** Thank you for comment. In general, two-way ANOVA is used to study the effect of two independent variable on a dependent variable. Moreover, the two-way ANOVA test is similar to the two-sample t-test, which was used for statistical analysis in the present study.
As suggested by the reviewer, we do agree and analyzed two independent factors (concentrations of tannic acid and types of drying method) on dependent variable (analytical parameter) using two-way ANOVA (GraphPad Software, San Diaego, CA USA) followed by Dunnett’s multiple comparison. The discussion based on the analysis has been included. Please see line 188-192. This discussion on the data was rechecked and revised in ‘Results and discussion’ part if needed.
In its current state, the level of English throughout your manuscript does not meet the journal's desired standard. There are some grammatical and spelling errors and full stops missing as well as instances of badly worded/constructed sentences. Please check the manuscript and refine the language carefully. I suggest that you should ask several colleagues who are skilled authors to check the English before your submission.
***** Thank you for your comment. The manuscript has been checked for typos and grammar mistakes using ‘Grammarly’ software. The corrections have been made in revised manuscript.
The main content of the abstract should include the brief purpose of the research, the principal result, and the major conclusion. The abstract, in the present form, is not adequate. Additionally, (i) the authors must state the revised justification in the abstract to support the study; (ii) some results should be nice.
***** Thank you for the suggestion. The comment has been well taken into consideration. The purpose of the research, the principal result, and the major conclusion have been added in the abstract. In addition, justification has been provided for better clarity. Please see line 21-24 and 35-36. However, for the results, the major results have been already given.
I have not found a sign that the appropriate statistical analysis has been done. When we say that a compound is greater than another, we must be supported by the statistical analysis, indicating the p-value (p <0.05 or p> 0.05)
***** Thank you for the suggestion. A significant difference indicating the p-value shown as (p <0.05) or (p >0.05) has been added in the revised manuscript.

Round 2
Reviewer 1 Report
The manuscript can be accepted at this version.
Reviewer 2 Report
Taking into account the responses to the comments and the revisions made, I consider that the manuscript has been sufficiently improved to be published in Foods in its current version.
Reviewer 3 Report
The paper has been improved. In the present form, the work is suitable for publication in Foods